# Study on the Effect of Two-Phase Anaerobic Co-Digestion of Rice Straw and Rural Sludge on Hydrogen and Methane Production

**Hengjun Tang [1], Cheng Tang [2], Heng Luo [3], Jun Wu [1], Jinliang Wu [4,\*], Jian Wang [5], Libo Jin [6] and Da Sun [6,\*]**

[1] School of Civil Engineering, Sichuan University of Light and Chemical Engineering, Zigong 643000, China; xingen2000@126.com (H.T.); wujun-st@foxmail.com (J.W.)

[2] School of Architecture and Engineering, Chongqing University of Science and Technology, Chongqing 401331, China; 2021206010@cqust.edu.cn

[3] Geological Research Institute of No. 9 Oil Production Plant of CNPC Changqing Oilfield, Yinchuan 750006, China; luoheng_cq@126.com

[4] Chongqing Architectural Design Institute Co., Ltd., Chongqing 400044, China

[5] Chongqing Yubei District Ecological Environment Monitoring Station, Chongqing 401124, China; kingj2000@163.com

[6] State and Local Joint Engineering Research Center for Ecological Treatment Technology of Urban Water Pollution, School of Life and Environmental Science, Wenzhou University, Wenzhou 325035, China; 20160121@wzu.edu.cn

\* Correspondence: wujliang@163.com (J.W.); sunday@wzu.edu.cn (D.S.); Tel./Fax: +86-186-8086-8992 (J.W.); +86-183-5878-2040 (D.S.)

**Abstract:** Hydrogen and methane, as chemical raw materials with broad application prospects in the future market, can be produced by the two-phase anaerobic co-digestion of rice straw and sludge. The study was carried out using a medium-temperature batch experiment with rice straw, a rural crop residue from Sichuan, and residual sludge from a sewage treatment station. The effect of the mixing ratio of rice straw and rural sludge on hydrogen and methane production from anaerobic digestion was investigated with a view to alleviating the energy crisis and efficient resource utilization. The experimental results showed that hydrogen production was most favorable when rice straw/sludge = 5:1, with a cumulative hydrogen yield as high as $38.59 \pm 1.12$ mL/g VSadded, while methane production was most favorable when 3:1, with a cumulative methane yield as high as $578.21 \pm 29.19$ mL/g VSadded. By calculating the energy yield, it was determined that 3:1 is more favorable for the two-phase anaerobic digestion capacity of rice straw and sludge, which is as high as $20.88 \pm 1.07$ kJ/g VSadded, and its conversion of hydrogen and methane is 0.75% and 78.19%, respectively. The hydrogen production pathway was dominated by the butyric acid type, whose hydrogen production phase pH ($5.84 \pm 0.13$) was slightly higher than the optimal pH for hydrogen-producing bacteria, while the methanogenic phase could meet the optimal pH for methanogenic bacteria ($6.93 \pm 0.17$).

**Keywords:** rice straw; rural sludge; two-phase anaerobic digestion; mixing ratio; hydrogen yield; methane yield



## 1. Introduction

China is the largest rice producer in the world, and a large amount of rice straw is left behind after rice harvest every year. In addition, with the rapid development of urbanization, the use of activated sludge technology for wastewater treatment will also generate a large amount of surplus sludge [1]. Since the 18th National Congress of the Communist Party of China, with the continuous improvement of environmental standards in response to President Xi Jinping's call for "green waters and green mountains", efforts need to be made to develop renewable resources [2], purify and utilize rice straw and sludge

through anaerobic co-digestion to produce hydrogen and methane, and provide an effective method for the treatment of agricultural solid waste [3]. Regarding resource utilization technologies for these two typical types of rural solid waste, Jinhe Jiang et al. [4] concluded that two-phase anaerobic digestion is better than single-phase anaerobic digestion for maximizing hydrogen and methane production from solid organic wastes, and has the advantage of better reaction stability. Two-phase anaerobic digestion can recover hydrogen and methane at the same time, and has the advantages of low energy consumption [5], sustainability, and high energy production. It is considered one of the most effective methods to cope with the energy crisis [6], which can realize the resourceful utilization of rural solid waste and generate clean energy, and has attracted the attention of scholars at home and abroad. Currently, research on two-phase anaerobic hydrogen and methane production using food waste is common [7,8], but there are few reports on hydrogen and methane production from agricultural (village) waste.

Rice straw and rural municipal sludge are two typical types of agricultural (village) solid waste, and their common treatment methods are open burning, direct return to farmland, and sanitary landfill, sludge incineration, sludge composting, and being sent to urban areas for centralized processing [9,10]. The disposal of municipal sludge has historically been a big issue. If not handled properly, it is easy to cause secondary environmental pollution and financial losses [11]. In recent years, with the continuous development of sludge treatment and disposal technologies, researchers have developed a number of new treatment technologies. For example, thermochemical transformations, hydrothermal liquefaction, etc. However, these new treatment technologies are difficult to apply on a large scale due to the constraints of technical means and treatment costs. Current sludge disposal methods are still dominated by stabilized landfills, aerobic fermentation, dry incineration, anaerobic digestion, and the use of building materials. On the other hand, China is a large country with a large population, and rice is one of the main staple foods in China, so China is rich in rice straw resources. Currently, the most common ways to reuse rice straw in China are directly returning it to the field and mixing it with manure for composting. Returning rice straw directly to the field is a simple, economical, and rapid method of reuse. However, as the quality of rice straw after returning to the field is low and it is difficult to spread evenly, it not only increases the difficulty of plowing, but also affects the seeds and emergence of crops to a certain extent. Although rice straw and sludge have good anaerobic resource utilization potential [12,13], their single digestion can lead to lower gas production rates due to factors such as an inappropriate C/N ratio. Conversely, co-digestion can significantly improve gas production rates due to its advantages of balancing nutrients, diluting toxic substances, and improving microbial activity [14]. Hairong Yuan et al. [15] explored the performance of the anaerobic co-digestion of sludge and wheat straw. The results of the study showed that the anaerobic digestion performance of sludge was affected by different C/N ratios, and the use of anaerobic co-digestion resulted in higher gas production than single sludge. Xuhui Wang [16] used mixed aerobic composting of rice straw and residual sludge to study the effect of C/N ratio and different particle sizes of straw on the treatment of composting system, and find the best C/N ratio. Jing Ning et al. [17] conducted a study on the gas production rate of anaerobic co-digestion using pig manure and corn stalk as substrates. The results showed that compared to single anaerobic digestion, when pig manure and corn stalk were co-digested at a C/N ratio of 25, the reaction system operated stably and had the highest specific gas production rate. Taotao Tang [18] found that sludge combined with corn stover (1:1.5) had a strong promotion of methane production using anaerobic digestion technology. Qiuying Huang [19] studied a co-fermentation method to improve the gas production performance of rice straw and obtained the same results. In order to construct an efficient and stable rice straw sludge with a combined anaerobic digestion system, Yiran Wang [20] conducted a batch experiment of combined anaerobic digestion using different types of rice straw and excess sludge. The research results showed that the addition of rice straw significantly increased the methane content compared to the single anaerobic digestion of sludge.

At present, anaerobic digestion using rice straw/sludge has limited methane production, but there is a lack of research reports on its co-digestion for hydrogen production and methane production. Therefore, this study uses a medium-temperature batch experiment to investigate the optimal mixing ratio of rice straw and rural municipal sludge for hydrogen and methane production in order to provide experimental support for its resourcefulness and provide reference for the low utilization rate of straw and sludge and the decline of land fertility in China.

## 2. Materials and Methods

### 2.1. Substrate and Inoculated Sludge

Rice straw was obtained from a rural area in Sichuan, naturally dried, ground into powder, passed through a 0.3 mm sieve, and stored at room temperature. The sludge was residual sludge from a rural sewage treatment station in Sichuan, which was mixed with a small amount of tap water and then passed through a 1 mm sieve to remove large particulate matter, and stored at 4 °C. Inoculated sludge was taken from a laboratory (VS = 2.25 ± 0.09 mg/g, TS = 4.45 ± 0.12 mg/g) and adjusted TS = 3%. In order to inhibit the activity of methanogens [21], the inoculated sludge pH = 2.0 ± 0.1 was adjusted using hydrochloric acid or sodium hydroxide solution, and then left at room temperature for 24 h. Rice straw and sludge were formulated according to 1:0, 5:1, 3:1, 1:1, and 0:1 (TS), with an initial TS = 3% for each mixed substrate. The properties of the mixed substrate are shown in Table 1.

**Table 1.** Characteristics of mixed substrates.

| Rice Straw/Sludge (TS-Based) | TCOD (g/L) | SCOD (g/L) | VS (g/L) | TS (%) | pH |
|---|---|---|---|---|---|
| 1:0 | 31.49 ± 2.94 | 5.34 ± 0.32 | 2.64 ± 0.09 | 3.00 ± 0.02 | 7.29 ± 0.23 |
| 5:1 | 34.53 ± 1.94 | 5.67 ± 0.27 | 2.51 ± 0.11 | 3.00 ± 0.01 | 7.19 ± 0.03 |
| 3:1 | 36.02 ± 1.53 | 5.95 ± 0.23 | 2.46 ± 0.12 | 3.00 ± 0.02 | 7.22 ± 0.05 |
| 1:1 | 37.94 ± 2.33 | 6.34 ± 0.17 | 2.41 ± 0.14 | 3.00 ± 0.03 | 7.35 ± 0.20 |
| 0:1 | 42.76 ± 3.23 | 7.43 ± 0.67 | 2.28 ± 0.06 | 3.00 ± 0.02 | 7.41 ± 0.18 |

### 2.2. Batch Test Device and Test Method

The batch test was conducted in a blue cap bottle with a total volume of 300 mL and a working volume of 200 mL. A rubber plug was used instead of the bottle cap. There was a liquid sampling port and gas collection port on the rubber plug, and the gas was collected through small air bags. The gas was collected in a small gas bag, as shown in Figure 1.

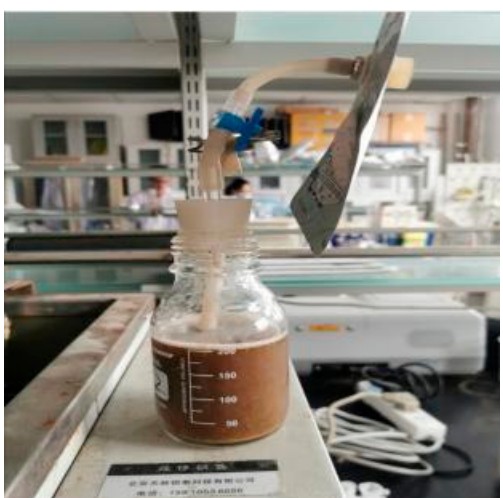

**Figure 1.** Experimental setup.

To perform the hydrogen production test, 150 mL of prepared mixed substrate and 50 mL of acid-treated inoculated sludge were first added to the blue-capped vials. The initial pH was adjusted to $7.0 \pm 0.1$, the vial was filled with nitrogen for 5 min and plugged with the rubber stopper, then the blue cap vials were placed in a water bath oscillator and the reaction temperature was set to $37 \pm 1$ °C and the oscillation rate to 120 rpm. Three sets of parallel experiments were set up for each mixed ratio of substrate. During the reaction process, the system pH was not controlled or adjusted, and the gas production and hydrogen content were measured regularly until the end of hydrogen production.

At the end of hydrogen production, the water bath oscillator was turned off, and 10 mL of liquid sample was taken through the liquid sampling port while shaking well. pH was promptly measured and the sample was stored at 4 °C for subsequent analysis. Each blue-capped bottle was then filled with 10 mL of unpretreated TS = 3% inoculated sludge, and the sampling port was closed after five minutes of nitrogen filling. The methanogenic reaction was carried out at a temperature of $37 \pm 1$ °C and an oscillation rate of 120 rpm. Measurements of gas production and methane content were taken at regular intervals until the end of methane production.

### 2.3. Analytical Methods

TS, VS, COD, and SCOD were measured using the APHA standard method [21]; pH was measured using a desktop pH meter (PHS-3E, Youke, Shanghai, China). Gas production was measured by pumping with a syringe. The contents of hydrogen, methane, and VFAs were determined using a gas chromatograph (6890 N, Agilent, Santa Clara, CA, USA) [22]. TDX-01 packing column and thermal conductivity detector (TCD) were also adopted for this. The chromatographic conditions were as follows: argon gas with a flow rate of 40 mL/min was used as the carrier gas, the inlet temperature was not controlled, the furnace temperature was 170 °C, and the detector temperature was 220 °C. The whole gas determination time was 2.5 min. Hydrogen and methane gas production, energy yield, COD removal, and equilibrium were calculated using Eq. [22].

## 3. Results and Analysis

### 3.1. Hydrogen and Methane Production

After two-phase anaerobic digestion, the organic substances in rice straw and sludge were recovered in the form of hydrogen and methane in the first and second stages, respectively. The hydrogen and methane yields can directly reflect the digestion performance. As shown in Figure 2A, the cumulative hydrogen yield in descending order was in the groups of rice straw/sludge = 5:1, 1:0, 3:1, 1:1, and 0:1, which is the same pattern as that of Chen et al. [14], who utilized rice straw and swine manure in anaerobic fermentation at high temperatures for hydrogen production. The difference is that the maximum cumulative hydrogen production rate ($38.59 \pm 1.12$ mL/g VSadded) in this article is slightly lower than that obtained by Hong Chen et al. [14] (44.59 mL/g VSadded), because complex organic substances (such as rice straw and sludge) are more easily utilized by microorganisms under high-temperature conditions.

After fermentation for hydrogen production, the cumulative methane yield from anaerobic digestion for each group is shown in Figure 2B. As can be seen from the figure, the methane yields of both rice straw and sludge co-digestion were greater than their respective mono-digestion yields. This suggests that co-digestion facilitates the balancing of substrate nutrients (e.g., C/N), thereby increasing microbial activity. In addition, co-digestion can also effectively dilute toxic substances contained or generated in a single substrate, such as S and N elements, which can generate $H_2S$ and $NH_4^+$-N [23] in anaerobic environments, which have strong toxic and side effects on microorganisms. When the mixing ratio was 3:1, the highest methane yield ($578.21 \pm 29.19$ mL/g VSadded) was obtained. Compared to Zhang et al. [24], who studied two-phase anaerobic methane production using rice straw and pig manure (188.79 mL/g VSadded), in this study, not only was a higher methane yield obtained, but hydrogen recovery was also realized.

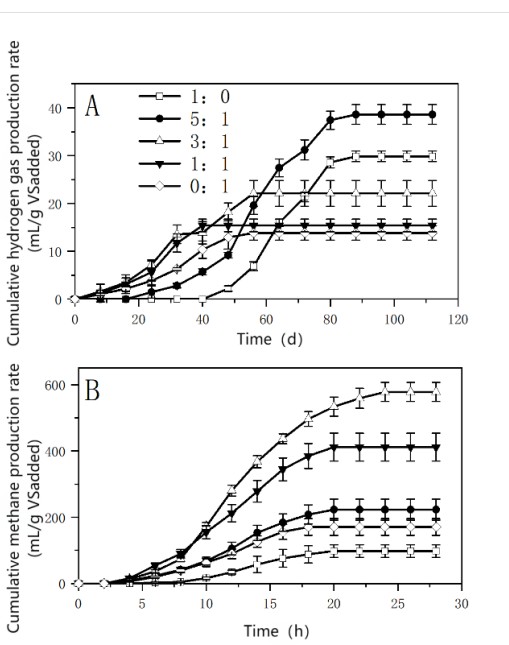

**Figure 2.** Accumulation of hydrogen (**A**) and methane (**B**) yield of rice straw and sludge under different mixing ratios.

The energy yield can be a good measurement of the two-phase anaerobic capacity of rice straw and sludge, and the results of energy yield calculation for each group are shown in Figure 3. As shown in the figure, methane is the main product at each mixing ratio, because most organic substances still exist in the fermentation broth in the form of VFAs during the hydrogen production stage. The hydrogen production time of fermentation is much less than the methane production time of digestion (Figure 2). The hydrogen and methane energy yields of each group were consistent with their cumulative hydrogen and methane yields, and the highest energy yield (20.88 ± 1.07 kJ/g VSadded) was obtained when rice straw/sludge = 3:1. Of these, hydrogen and methane accounted for 1.15% and 98.85%, respectively. The results of Rafieenia et al. [8] for the total energy yield (21.69 kJ/g VSadded) using food waste for two-phase anaerobic hydrogen production (21.kJ/g VSadded) are close to the results of this paper.

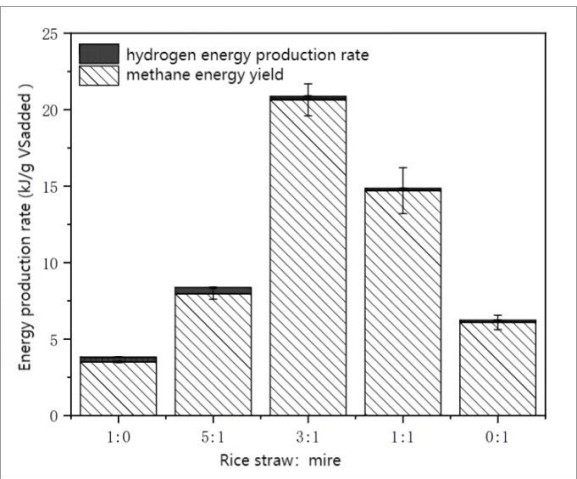

**Figure 3.** Energy yield of rice straw and sludge at different mixing ratios.

### 3.2. VFAs and pH

After hydrogen production, the concentration of VFAs was positively correlated with the hydrogen yield, i.e., the highest VFAs concentration (1539.19 $\pm$ 87.26 mg/L) was obtained when rice straw/sludge = 5:1, indicating that the activity of hydrogen-producing bacteria is the highest at this ratio. At each mixing ratio, butyric acid is the most important VFA, and has the same pattern as the hydrogen yield. That is, when rice straw/sludge = 5:1, the highest butyric acid concentration (1113.31 $\pm$ 65.51 mg/L) is obtained, and when the ratio is 0:1, the lowest butyric acid concentration (190.11 $\pm$ 13.01 mg/L) is obtained. This indicates that the hydrogen production type of rice straw and sludge is the butyric acid type, and Hong Chen et al. (2020) [14] also obtained the same type using rice straw and pig manure for hydrogen production under high temperatures.

During the methanogenic stage, methanogens utilize VFAs to generate methane, resulting in a significant decrease in all component VFAs. Kanokwan et al. [25] concluded that the concentration of VFAs during methanogenesis should not be higher than 400 mg/L to ensure that H+ does not inhibit methanogenic bacteria. However, when rice straw/sludge = 1:0, the VFA concentration after methanogenesis is 560.51 $\pm$ 11.83 mg/L, indicating that the mixing ratio does not create a suitable environment for methanogenic bacteria to survive. Meanwhile, the concentration of VFAs in the remaining fractions was below 400 mg/L. This result corresponds to its cumulative methane production rate (Figure 2).

The methane production process consumes acidic VFAS, so the pH of individual components after methane production is higher than that after hydrogen production (Figure 4). Both the pH after hydrogen production and the pH after methane production increase with the increase in the sludge content in the substrate, which is due to the hydrolysis of NH4$^+$-N generated by the N-rich element in the sludge, thereby increasing the pH. Secondly, the optimal pH for the hydrogen production process and the methane production process is 5.0–5.5 [26] and 6.5–8.2 [27], respectively. This indicates that the optimal pH for the hydrogen production process can be met only when rice straw/sludge = 5:1. Below this pH range, at rice straw/sludge = 1:0 (4.84 $\pm$ 0.04), conditions were not suitable for hydrogen production. The pH was slightly higher than the optimum pH (5.84 $\pm$ 0.13) for hydrogen-producing bacteria at rice straw/sludge = 3:1, and the rest of the ratios are above 6.0 pH. For methanogenic processes, when the mixing ratio is 3:1, 1:1, and 0:1, all of them can satisfy the optimum pH of the methanogenic process, 6.93 $\pm$ 0.17, 7.34 $\pm$ 0.43, and 7.66 $\pm$ 0.14, respectively. At 5:1, the pH (6.43 $\pm$ 0.29) is lower because of less methane production (Figure 2).

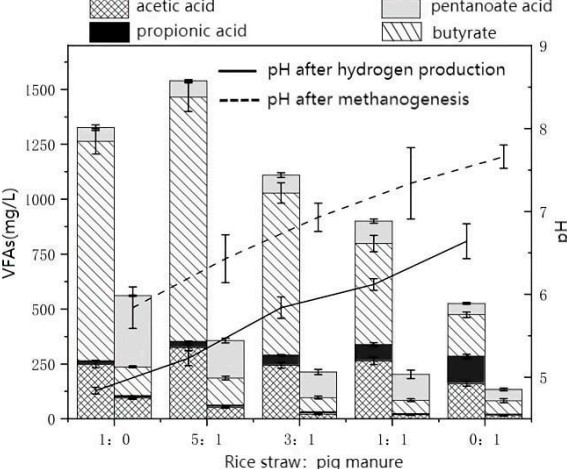

**Figure 4.** The VFAs and pH after hydrogen production and methane production. Note: the first column under the same mixing ratio is the VFAs after hydrogen production, and the latter column is the VFAs after the methane production.

### 3.3. COD Removal Rate and Material Balance

The COD removal rates of rice straw and sludge during single digestion were 33.37% and 37.62%, respectively. It was further shown that two-phase anaerobic co-digestion favors the enhancement of microbial activity. The highest COD removal of 80.20 ± 5.20% was recorded at rice straw/sludge = 3:1. Chen et al. [28], in their study of two-phase anaerobic methane production from rice straw and swine manure (44.59–56.93%), showed lower COD removal than in this paper. This may be due to the comparison to two-phase anaerobic methane production and the fact that two-phase anaerobic hydrogen and methane production provide optimal environments for hydrogen-producing and methanogenic bacteria, respectively, thereby dramatically increasing microbial activity and leading to more favorable COD removal. In contrast, Qin et al. [29] obtained 83.00% COD removal using food waste and waste paper for two-phase anaerobic hydrogen methanogenesis (Figure 5).

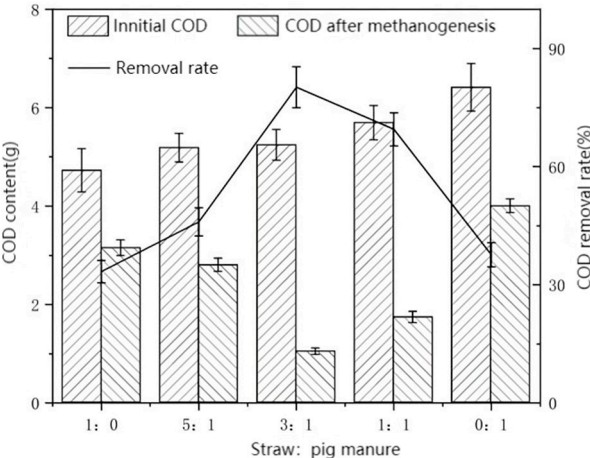

**Figure 5.** TCOD removal efficiency at different rice straw and sludge ratios.

Material balance can directly reflect the transformation of organic substances in the substrate after two-phase anaerobic digestion. As shown in Figure 6, solid COD (PCOD) accounts for a large proportion, which is due to the low biodegradability of rice straw and sludge, making it difficult for it to be utilized by hydrogen and methane producing bacteria. The highest hydrogen conversion (1.34%) was obtained when rice straw/sludge = 5:1, while the remaining groups decreased with their rice straw content; of these, the 0:1 group has a hydrogen conversion rate of only 0.35%. This is because compared to sludge, substances such as cellulose in rice straw have better hydrogen production potential. Theoretically, hydrogen can be converted at a rate of 7.5–15% [30], but due to the difficultly of degrading mixed substrates and differences in handling, the actual value is much lower than the theoretical value. For example, Abreu et al. [31] utilized garden waste for anaerobic hydrogen production with a hydrogen percentage of 1–3%.

The conversion rate of methane is much greater than that of hydrogen, which is similar to other studies [7,29]. The proportion of hydrogen and methane is related to the reaction time. In this study, the hydrogen production stage of each component was completely finished within 112 h, while the methane production stage lasted for 28 d, resulting in a lower proportion of hydrogen (0.35–1.34%) and a higher proportion of methane (15.67–78.19%). The highest methane conversion rate (78.19% with 0.75% hydrogen conversion) was obtained in the group of rice straw/sludge = 3:1, while the 5:1 group obtained the highest hydrogen conversion (30.88% and methane conversion of 1.34%). This suggests that different pairs of substrate-mixing ratios affect hydrogen and methane production during hydrogen and methane production.

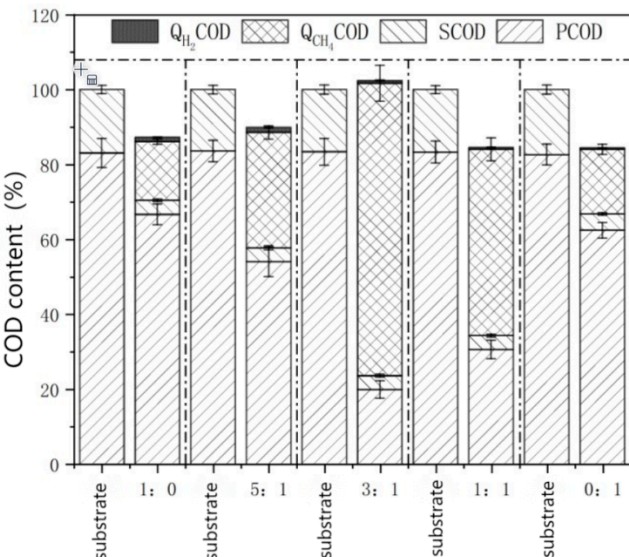

**Figure 6.** Material balance at different rice straw and sludge ratios.

## 4. Discussion

### 4.1. Optimization of Two-Phase Anaerobic Reaction Conditions

Based on the experimental results of two-phase anaerobic hydrogen and methane production from rice straw and rural sludge, the reaction conditions of two-phase anaerobic fermentation for hydrogen and methane production were optimized under different material ratios. After two-phase anaerobic digestion, the length of the reaction time is a decisive factor influencing the conversion of organic matter in the remaining substrate. This is due to the fact that rice straw and sludge are less likely to be utilized by hydrogen-producing and methanogenic bacteria, so a methane conversion rate greater than the hydrogen conversion rate occurs. This results in different requirements for substrate mixing ratios in the two-phase anaerobic digestion process. The 5:1 group obtained the highest hydrogen conversion (30.88% and methane conversion of 1.34%). The highest methane conversion (78.19% and hydrogen conversion of 0.75%) was obtained in the rice straw/sludge = 3:1 group. In addition, the material mixing ratio affects the activity of hydrogen-producing and methanogenic bacteria, which in turn affects the concentration of hydrogen and methane produced. That is, the highest concentration of VFAs (1539.19 ± 87.26 mg/L) was observed at rice straw/sludge = 5:1, hydrogen-producing bacteria had the highest activity, and the butyric acid concentration (1113.31 ± 65.51 mg/L) was the highest. Finally, the ratio of materials for hydrogen and methane production under two-phase anaerobic reaction is affected by pH. The optimum pH (5.29 ± 0.09) for the hydrogen production process can be satisfied when rice straw/sludge = 5:1. Due to less methane production, the pH (6.43 ± 0.29) was lower at rice straw/sludge = 5:1.

### 4.2. Increased Efficiency of Hydrogen and Methane Production

Based on the COD removal rate and material balance, two-phase anaerobic digestion can produce much more hydrogen and methane compared with single-phase anaerobic digestion. The two phases of anaerobic fermentation for hydrogen and methane production are instigated mainly through the separation of hydrogen and methane to improve the gas production rate of the whole anaerobic reaction. Liu et al. [32] used sludge and corn stover to conduct a mixed static fermentation test, and the experimental results showed that the mixed fermentation of the two is conducive to increasing the rate of hydrogen production, as well as the hydrogen and methane production rate. When the substrate concentration was 10 g-VS/L and the TS ratio of straw to sludge was 2:1, the hydrogen and methane yields of mixed fermentation reached 13.4 mL/g-VS and 172.6 mL/g-VS, respectively, which were 13 and 1.4 times higher than those of sludge alone. Cheng et al. [33] used alkali-

pretreated straw, and then anaerobically fermented it to produce hydrogen and methane, and the hydrogen yield was 58.0 mL/g-straw and the methane yield was 200.9 mL/g-straw, with 67.1% energy recovery. Zhu et al. [34] used a CSTR reactor for the anaerobic fermentation of potato waste. The hydrogen yield was 30 mL/g-VS, and the methane yield was 183 mL/g-VS. Figure 2B shows the cumulative methane yield of each group after fermentation for hydrogen production. As can be seen from the figure, the methane yield from mono-digestion of each ratio of rice straw and sludge was lower than that of co-digestion. Zhang et al. [24] obtained a low methane yield using two-phase anaerobic methanogenesis (188.79 mL/g VSadded) using rice straw and pig manure, whereas a higher methane yield (578.21 ± 29.19 mL/g VSadded) was obtained through the co-digestion of rice straw and sludge.

Currently, most of the material choices for the two-phase anaerobic digestion process for hydrogen and methane production are straw, livestock manure, sludge, and kitchen waste. The research on two-phase anaerobic co-digestion with straw and sludge as materials is still relatively scant. In this experiment, the two-phase anaerobic digestion of rural rice straw and sludge was used to study hydrogen and methane production. The ratios of material substrates that are be suitable for hydrogen-producing and methanogenic bacteria optimal values were determined. However, there is a lack of research on the two-phase anaerobic digestion of corn stover, wheat straw, and sorghum stover with sludge.

## 5. Conclusions and Outlook

The two-phase anaerobic co-digestion of rice straw and rural municipal sludge has a good performance in terms of hydrogen and methane production. The highest hydrogen yield (38.59 ± 1.12 mL/g VSadded) was obtained when rice straw/sludge = 5:1. The highest methane yield (578.21 ± 29.19 mL/g VSadded) was obtained when rice straw/sludge = 3:1. Due to the difference in reaction time, methane is the main energy source for two-phase anaerobic digestion, with relatively little hydrogen. The highest anaerobic capacity of rice straw and sludge (20.88 ± 1.07 kJ/g VSadded) was observed when rice straw/sludge = 3:1, at which the proportions of hydrogen and methane were 1.15% and 98.85%, respectively. When rice straw/sludge = 3:1, the pH (5.84 ± 0.13) of the hydrogen production stage was slightly higher than the optimum value of hydrogen-producing bacteria. And, the pH (6.93 ± 0.17) of the methanogenic stage can satisfy the optimum value of methanogenic bacteria.

In today's society, rice straw and rural municipal sludge are two common organic solid wastes. Anaerobic digestion is a cost-effective and environmentally promising method of organic waste management. On the one hand, the raw materials come from various organic wastes, which can be recovered as biomass energy and solve the problem of organic waste treatment and disposal. On the other hand, biogas, a clean and renewable energy source, can be produced, thus alleviating the fossil energy crisis. In two-phase anaerobic digestion systems, straw with a complex lignocellulosic matrix structure can result in low hydrogen production. Therefore, there are fewer studies on the anaerobic digestion of straw as feedstock for hydrogen and methane production. The use of rice straw as a raw material for anaerobic digestion for hydrogen and methane production has high environmental benefits. It can reduce solid waste and generate clean energy after fermentation, which can solve the problems of energy scarcity and environmental pollution. It is a sustainable technology. In the context of environmental ecology, in line with the trend of co-digestive technology of the times, anaerobic digestion can better realize the waste-to-wealth philosophy and the recycling of renewable resources, and provide the basic theoretical and technological support for their practical application.

**Author Contributions:** Conceptualization, H.T. and C.T.; methodology, H.L.; software, J.W. (Jun Wu); validation, J.W. (Jinliang Wu) and L.J.; formal analysis, J.W. (Jian Wang), H.L. and D.S.; surveys, D.S. All authors have read and agreed to the published version of the manuscript.

**Funding:** This research was supported by the Project of Key Laboratory of Synergistic Control and Joint Remediation of Water and Soil Pollution of National Environmental Protection (GHBK-2020-002); the Project of Sichuan Provincial Science and Technology Department (2021YJ0342); the Key Research and Development Project of Luzhou Municipal Bureau of Science, Technology and Talent Work (2020-SYF-20); the Postgraduate Innovation Program of Chongqing University of Science and Technology (No. YKJCX2220606); and the Energy Engineering Mechanics and Disaster Prevention and Mitigation Chongqing Key Laboratory (No. EEMDPM2021204).

**Institutional Review Board Statement:** Not applicable.

**Informed Consent Statement:** Not applicable.

**Data Availability Statement:** Data are contained within the article.

**Conflicts of Interest:** Author Wu Jinliang is an employee of Chongqing Architectural Design Institute Co., Ltd. No conflict of interest between Wu Jinliang and all authors.

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
