# Peer review of "Study on the Effect of Two-Phase Anaerobic Co-Digestion of Rice Straw and Rural Sludge on Hydrogen and Methane Production"

_sustainability, doi:10.3390/su152216112_

Round 1
Reviewer 1 Report
Comments and Suggestions for Authors
The manuscript presents the obtaining of hydrogen and methane by varying the proportions of rice straw and sludge in a two-phase anaerobic digestor. The results show the importance of that proportion for the preferential obtaining of either hydrogen or methane, as well as the changes in pH and COD. The latter is very remarkable to recognize if anaerobic treatment is efficient for removing contaminants from water.
The work is outstanding, and I would like to indicate some minor changes which I consider could improve the quality of the publication.
Authors should pay more attention to the manuscript text to correct flaws.
For instance, the authors should correct the chemical formula.
Please separate the number and its units.
It is worth highlighting the superscripts and subscripts.
Reviewer 2 Report
Comments and Suggestions for Authors
Regarding the experimental design, I have some questions to ask the authors:
· In Table 1 they present the mixtures they prepared to carry out the co-digestion experiments: Why did they only consider the 5:1 Rice straw: sludge ratio? I think that for the results to be representative, it would be necessary to include at least a ratio intermediate between 1:1 and 5:1, and at least a ratio in which the amount of sludge is greater than that of rice straw.
· In item 2.3, the authors mention “The content of hydrogen, methane, and VFAs was 129 determined using a gas chromatograph (6890N, Agilent, USA) [22].” It would be interesting if they described the analytical methodology a little more, at least the type of column and detector they used, and how they determined the response factors to quantify hydrogen, methane, VFAs.
· Why did they not quantify other gases such as H2S?
Results
Figure 3 shows the 3:1 ratio, which offers the best energy production rate. However, this is not an experiment of those proposed in Table 1. How is this possible?
The same happens in Figure 2, 4, 5
Discussion
On line 282, the authors mention the 2:1 Rice straw: sludge ratio, but it is not one of the experiments in Table 1 nor do results of this relationship appear in the results section. Here there is inconsistency between the experiments carried out, the results shown and the discussion generated from them.
Reviewer 3 Report
Comments and Suggestions for Authors
Effects of rice straw/rural sludge mass ratio on the performance of anaerobic co-digestion process are presented in the paper. The subject is topical in the related field, but the text should be very carefully revised. There are many mistakes and redundant information. I suggested some modifications in the attached document.

I think it is necessary to have the manuscript reviewed by a native English speaker.
Author Response
Changes are marked,in the box

Round 2
Reviewer 2 Report
Comments and Suggestions for Authors
I consider that the manuscript has been substantially improved, thanks to the fact that the authors modified all the requested questions.
I recommend the publication of the paper in the journal
Reviewer 3 Report
Comments and Suggestions for Authors
I think the paper can be published in this form.